**Data Availability Statement:** All the relevant data were included in the manuscript. However, it is

# Individual and community-level determinants of knowledge of ovulatory cycle among women of childbearing age in Ethiopia: A multilevel analysis based on 2016 Ethiopian Demographic and Health Survey

**Baye Dagnew**[1]*, **Achamyeleh Birhanu Teshale**[2], **Henok Dagne**[3], **Mengistie Diress**[1], **Getayeneh Antehunegn Tesema**[2], **Reta Dewau**[4], **Meseret Derbew Molla**[5], **Yigizie Yeshaw**[1,2]

1 Department of Human Physiology, School of Medicine, College of Medicine and Health Sciences, University of Gondar, Gondar, Ethiopia, 2 Department of Biostatistics and Epidemiology, Institute of Public Health, University of Gondar, Gondar, Ethiopia, 3 Department of Environmental and Occupational Health and Safety, Institute of Public Health, University of Gondar, Gondar, Ethiopia, 4 Department of Epidemiology and Biostatistics, School of Public Health, College of Medicine and Health Sciences, Wollo University, Dessie, Ethiopia, 5 Department of Biochemistry, School of Medicine, College of Medicine and Health Sciences, University of Gondar, Gondar, Ethiopia

* bayedagnew7@gmail.com

## Abstract

### Background

Knowledge of the ovulatory cycle (KOC) aids women to refrain and engage in sexual intercourse to avoid and to get pregnancy, respectively. The effect of community-level factors on KOC was not yet known in Ethiopia. Therefore, we aimed to investigate the community- and individual-level determinants of KOC among women of childbearing age.

### Methods

We used the 2016 Ethiopian Demographic and Health Survey, and total weighted samples of 15,683 women were included. Intra-class correlation, median odds ratio, and deviance were executed for model comparison in which a model with the lowest deviance was the best model i.e. model III in this case. A multivariable multilevel logistic regression model was employed to identify community- and individual-level factors of correct KOC. In the ultimate model, an adjusted odds ratio (AOR) with a 95% confidence interval was reported and variables with a p<0.05 were considered as statistically significant.

### Results

In this study, 3,698 [23.58% (95% CI; 22.92–24.25)] participants had correct KOC. Women's age in years, i.e. 20–24 (AOR = 1.46;1.28–1.68) 25–29 (AOR = 1.72; 1.49–1.99), 30–34 (AOR = 2.21; 1.89–2.58), 35–39 (AOR = 1.78; 1.51–2.09), 40–44 (AOR = 1.97; 1.65–2.37), and 45–49 (AOR = 1.78; 1.44–2.19), knowledge of contraceptive methods (AOR =

ethically not acceptable to share the DHS data set with third parties and anyone who wants the data set can access the Measure DHS program at www.dhsprogram.com, through legal requesting. The authors had no special access privileges others would not have. We used the Ethiopian demographic and health survey (https://dhsprogram.com/methodology/survey/survey-display-478.cfm).

**Funding:** The authors received no specific funding for this work.

**Competing interests:** The authors declared that no competing interests exist.

3.08; 2.07–4.58), increased women's educational level, i.e. higher (AOR = 4.24; 3.54–5.07), secondary (AOR = 2.89; 2.48–3.36), and primary (AOR = 1.57; 1.39–1.78), higher household's wealth index, i.e. richest (AOR = 1.71; 1.35–2.16), richer (AOR = 1.42; 1.16–1.72), middle (AOR = 1.29; 1.07–1.56), and poorer (AOR = 1.24; 1.03–1.48), current contraceptive use (AOR = 1.26; 1.13–1.39), menstruating in the last six weeks (AOR = 1.13; 1.03–1.24), women's media exposure (AOR = 1.20; 1.07–1.35), and being in the community with a high level of media exposure (AOR = 1.53; 1.24–1.88) were statistically significant with KOC.

## Conclusions

Knowledge of the ovulatory cycle was low in this study, which demands health education for women of childbearing age. Special attention should be given to teenagers, those with lower educational, and lower economic status. Besides, the strengthening of media campaigns could increase women's KOC, which is crucial for preventing unintended pregnancy.

## Background

Ovulation is a physiological event noted by the break-up and exit of the dominant follicle from the ovary into the fallopian tube for potential fertilization [1]. The time of ovulation is detected by the knowledge of the basal body temperature and cervical mucus [2]. Understanding of ovulation helps women to conceive a child or escape sexual contact in the fertile period for the sake of contraception [3]. Ovulatory cycle is one of the fertility awareness family planning methods where people use measurements or on-time changes of the body to decide for sexual intercourse on a fertile period [4,5]. It is also useful to recognize certain pathologies [5], evaluate woman's health [6], and monitor fertility [7] whereas inadequate KOC predisposes to unintended pregnancy [8].

Evidences showed the prevalence of KOC was 32.8% in USA [8], 31.2% in Spain [9], and 15% in India [10]. In Africa, the prevalence of KOC ranged from 10.4%-49% such as studies in Togo (42.8%) [11,12]. Besides, 38% of women of childbearing age reported correct KOC in Ghana [13]. Few pieces of the literature disclosed the association of KOC with low socioeconomic status and encountering unwanted pregnancy [14], age (younger women had inadequate KOC) [15,16], education [14,17–19], and unintended pregnancy [11,16].

Even though KOC is a fundamental aspect of the female reproductive cycle, there is limited study in Ethiopia. Only one study reported the prevalence of knowledge of the ovulation period (i.e. 23.6%) which used the Ethiopian Demographic and Health Survey (EDHS) 2016 [20]. The previous study did not consider the community-level factors (only the individual-level factors were analyzed; even they did not use multilevel analysis which is appropriate for hierarchical data), there was no weighting to account for survey design, and essential variables were missed such as media exposure that plays a key role for the provision of information. Besides, the individual-level factors, knowledge of community-level factors aids to institute interventional strategies. Therefore, this study aimed to identify individual and community-level determinants of correct KOC among women of childbearing age in Ethiopia using the EDHS 2016. The results will be helpful to establish reproductive health specific interventions to advocate appropriate KOC. This will improve women's health to prevent unintended pregnancy especially for those with poor access and awareness of contraceptive methods,

particularly in Sub-Saharan Africa [21]. Forthcoming researchers will get baseline information for further studies on the field.

## Methods data source, study design, and study period

We used the latest EDHS (2016) data after a reasonable request from the Measure DHS programme [22] available at (https://dhsprogram.com/Data/terms-of-use.cfm). The Central Statistical Agency (CSA) had surveyed the data in collaboration with the Ethiopian Ministry of Health and Ethiopian Public health institute and also the international classification of functioning, disability and health (ICF) involved in technical assistance. Financial support was obtained from the development partners i.e. United States Agency for International Development (USAID), government of the Netherlands, Global Fund, Irish Aid, World Bank, United Nations Population Fund (UNFPA), United Nations Children's Fund (UNICEF), and UN Women. The dataset comprised all females aged 15–49 years. This is important data to provide the key health indicators at the national and sub-national level [23]. We used the Ethiopia Population and Housing Census (2007) as a sampling frame for the EDHS 2016 which was a complete list of 84,915 enumeration areas (EAs) created for the 2007 PHC. An EA is a geographic area covering an average of 181 households (~15,369,615 households on average). Each large EA was segmented to minimize the task of household listing, from which only one segment was selected for the survey, with probability proportional allocation to segment size. Household listing was conducted only in the selected segment, i.e. a 2016 EDHS cluster is an EA or a segment of an EA. Then, 18,008 households were selected for the sample, of which 17,067 were occupied. Of the occupied households, 16,650 were successfully interviewed, yielding a response rate of ~98%. Regarding the participants, 15,683 (15–49 years of age) were females and 12,688 (15–59 years of age) were males. The EDHS survey 2016 was a community-based cross-sectional study, and the data was collected from January 18, 2016 to June 27, 2016 [24].

### Study area

This survey was conducted in Ethiopia, one of the low-income countries in East Africa with 3˚-14˚ N and 33˚–48˚ E. For administrative purpose, Ethiopia is divided into nine regional states (Afar, Amhara, Benishangul-Gumuz, Gambela Peoples region, Oromia, Harari, Southern Nations, Nationalities and People's (SNNP), Somali, and Tigray), and two city administrations (Addis Ababa and Dire Dawa).

### Population and sample size

The source population was all women of childbearing age (15–49 years old) living in Ethiopia and those living in the selected enumeration areas (EAs) served as the study population. In the selected households, all women who spent the night before the survey day were included in the study. We used a women's dataset (IR file) and 15,683 (weighted) women of childbearing age were used for the final analysis.

### Sampling procedure

The EDHS 2016 survey used a two-stage stratified cluster sampling technique to get the study participants. The stratification was performed using separate regions (the nine regions and Dire Dawa) into rural and urban areas; although Addis Ababa is entirely urban. Then each stratum was divided into clusters or EAs, comprised 200–300 households. In the first stage of sampling, 645 EAs were selected (202 from urban and 443 from rural) and in the second stage, a fixed number of 28 households per cluster were selected randomly from the list of

households (~15,369,615 households). The numbers of women interviewed and responded for the outcome variable (KOC) were 15,683 and hence there were no missing cases for the outcome variable.

### Data collection

After delivering training about the purpose of the survey, nationally recruited data collectors collected the data using a pretested, structured, and interviewer-administered questionnaire [23]. The questionnaire was prepared in English and then translated into the local languages.

### Study variables

The outcome variable was women's correct KOC. The respondents were asked, "when is the ovulation time?" with six responses designated as "during her period", "after period ended", "middle of the cycle", "before the period begins", "at any time", and "don't know" [25]. We performed recoding of the variables so that the outcome variable becomes binary (dichotomized) which is useful to execute regression. For this, we created a new variable called correct knowledge of the ovulatory cycle whereby the alternative "middle of the cycle was considered as correct KOC and recoded as 1" and others were recoded as 0". The independent variables include individual-level factors (age, wealth index of the household, women's media exposure, educational status of the women, region, religion, marital status) [14–19] and community-level factors; non-aggregated (residence) and aggregated (community-level media exposure, community education level, and community poverty level). Community-level factors were created at cluster level using their respective individual-level factors to assess their effect at the community or cluster level.

Community-level media exposure was created from the women's exposure to radio, newspaper/magazine, and television (after merging these variables and recoding into Yes and No). Then, we took the proportion of women who had exposure to at least one of these media and categorized into low (if <50% of women had exposure to at least one media) and high (if ≥50% of women had exposure to at least one media) community-level media exposure using national median value since the data is not normally distributed. Community poverty level was created from household wealth index by recoding poorest and poor as poor. Then, the proportion of women from households with poor household wealth index was calculated and categorized as low poverty level (those with ≥ 50%) and higher poverty level (those with <50%) using national median value. Community-level education was created by aggregating the individual level woman's education at cluster level by taking the proportion of women with no education, which was similar as performed for the wealth index. Then we categorized it as a low and higher level of community education using a national median value like media exposure.

### Data management and statistical analysis

Stata version 16 was used for extraction, recoding, and analyzing the data. Before any statistical procedure, sampling weight was executed (using women's sampling weight) to account for the non-proportional allocation of the sample in each region and the potential variations in the response rates, as well as to get an appropriate statistical estimate. As described, stratification was performed to account for survey design. Tabulation, graphical, and textual presentations were used for descriptive results. We estimated the clustering effect since the DHS data has a hierarchical nature (random-effects) using intra-class correlation coefficient (ICC) and median odds ratio (MOR), which were used to check the need for the advanced models such as multilevel logistic regression analysis. We fitted four models; null model/ (containing only

the outcome variable), model I (individual-level factors only), model II (community-level factors only), and model III (the individual- and community-level factors simultaneously). Model III was the best-fitted model for this data since it had the lowest deviance value compared to the others. A multilevel binary logistic regression analysis was performed to identify individual and community-level determinant factors of KOC. Variables with a p-value <0.2 in the bi-variable multilevel regression were considered for the multilevel multivariable logistic regression analysis. Independent factors with a p<0.05 were considered as statistically significant factors of correct KOC. The adjusted odds ratio (AOR) along with its 95% confidence interval (CI) was reported to show the strength and direction of the association between correct KOC and independent factors.

### Ethical considerations

We got online permission to access the data from the DHS program, and the dataset was allowed for public use. During the survey, all the ethical issues were secured to ascertain confidentiality.

## Results

### Sociodemographic profiles of the participants

A total weighted samples of 15,683 women were included for the analysis. About one-fifth (21.56%) were aged 15–19 years with a median age of 27 (IQR = 20–35). Most respondents were from Oromia region (36.35%), Orthodox Christian followers (43.27%), rural dwellers (77.84%), and currently married (65.19%). Of the total participants, 47.81% did not attend formal education and 5.59% achieved higher education. Moreover, 26.55% and 56.07% were from the richest households and had no media exposure, respectively. In terms of contraceptives, 98.32% of women had self-reported knowledge of contraceptive methods, whereas 25.34% of women used contraceptives currently (Table 1).

### Knowledge of the ovulatory cycle among women of childbearing age

From the total of 15,683 women of childbearing age, 3,698 [23.58% (95% CI: 22.92–24.25)] had correct KOC. Those who replied to the question "what is the ovulation time?" as "after period ended" were 24.94% and 19.74% did not know the ovulation time when to occur. When performed as per the correct knowledge and incorrect knowledge, 11,985 (76.42%) women had incorrect knowledge (Fig 1).

### Determinants of KOC among women of childbearing age in Ethiopia

After running the bi-variable multilevel logistic regression analysis, the candidate variables for multivariable multilevel analysis (p<0.2) were the age of women in years, knowledge of contraceptive methods, educational status of the women, household's wealth index, current use of contraceptives, menstruation in the last six weeks, women's media exposure status, residence, community-level poverty, community-level media exposure, and community-level of women's education. In the multivariable multilevel logistic regression, all the aforementioned individual-level factors and one community-level factor (community-level media exposure) were significantly associated with correct KOC at p<0.05.

The odds of KOC among women aged 20–24, 25–29, 30–34, 35–39, 40–44, and 45–49 were 1.46 (95% CI = 1.28–1.68), 1.72 (95% CI = 1.49–1.99), 2.21 (95% CI = 1.89–2.58), 1.78 (95% CI = 1.51–2.09), 1.97 (95% CI = 1.65–2.37), and 1.78 (95% CI = 1.44–2.19) times compared to those aged 15–19 years, respectively. The odds of correct KOC was 3 times (95% CI = 2.07–

**Table 1. Sociodemographic profiles of the participants on knowledge of the ovulatory cycle in Ethiopia (n = 15683).**

| Variables | Categories | Weighted frequency (%) | Correct KOC frequency (%) |
|---|---|---|---|
| **Age** | 15–19 | 3,381 (21.56) | 685 (20.26) |
| | 20–24 | 2,762 (17.61) | 752 (27.240) |
| | 25–29 | 2,957 (18.85) | 796 (26.91) |
| | 30–34 | 2,345 (14.95) | 593 (25.31) |
| | 35–39 | 1,932 (2.32) | 401 (20.78) |
| | 40–44 | 1,289 (8.22) | 268 (20.75) |
| | 45–49 | 1,017 (6.48) | 685 (20.26) |
| **Region** | Tigray | 1,129 (7.20) | 199 (5.38) |
| | Afar | 128 (0.82) | 17 (0.47) |
| | Amhara | 3,714 (23.68) | 620 (16.77) |
| | Oromia | 5,701 (36.35) | 1783 (48.21) |
| | Somali | 460 (2.93) | 59 (1.60) |
| | Benishangul-Gumuz | 161 (1.02) | 150.42) |
| | SNNPR | 3,288 (20.97) | 532 (14.40) |
| | Gambela Peoples region | 44 (0.28) | 10 (0.26) |
| | Harari | 39 (0.25) | 16 (0.43) |
| | Addis Ababa | 930 (5.93) | 414 (11.22) |
| | Dire Dawa | 90 (0.58) | 32 (0.85) |
| **Religion** | Orthodox | 6,786 (43.27) | 1672 (45.21) |
| | Catholic | 120 (0.76) | 40 (1.07) |
| | Protestant | 3,674 (23.43) | 840 (22.72) |
| | Muslim | 4,893 (31.20) | 1097 (29.67) |
| | Others | 210 (1.34) | 49 (1.32) |
| **Marital status** | Never in Union | 4,037 (25.74) | 1018 (27.53) |
| | Currently married | 10,223 (65.19) | 2348 (63.48) |
| | Formerly married | 1,423 (9.08) | 333 (8.99) |
| **Educational status of women** | No education | 7,498 (47.81) | 1,231 (16.41) |
| | Primary | 5,490 (35.01) | 1,307 (23.81) |
| | Secondary | 1,818 (11.59) | 693 (38.12) |
| | Higher | 877 (5.59) | 468 (53.33) |
| **Wealth index** | Poorest | 2,633 (16.79) | 395 (14.99) |
| | Poorer | 2,809 (17.91) | 506 (12.0) |
| | Middle | 2,978 (18.99) | 557 (18.69) |
| | Richer | 3,100 (19.76) | 677 (21.86) |
| | Richest | 4,163 (26.55) | 1,564 (37.56) |
| **Women's media exposure** | No | 8,793 (56.07) | 1451 (39.23) |
| | Yes | 6,890 (43.93) | 2247 (60.77) |
| **Residence** | Urban | 3,476 (22.16) | 1300 (37.41) |
| | Rural | 12,207 (77.84) | 2,398 (19.64) |
| **Community media exposure level** | Low | 7,152 (45.60) | 1424 (38.50) |
| | High | 8,531 (54.40) | 2274 (61.50) |
| **Community education level** | Low | 6,754 (43.07) | 1677 (45.35) |
| | High | 8,929 (56.93) | 2021 (54.65) |
| **Community poverty level** | Low | 7,714 (49.18) | 145 (39.34) |
| | High | 7,969 (50.82) | 2243 (60.66) |
| **Knowledge of contraceptive methods** | No | 264 (1.68) | 14 (0.39) |
| | Yes | 15,419 (98.32) | 3684 (99.61) |
| **Current contraceptive status** | No | 11,709 (74.66) | 2,632 (22.48) |
| | Yes | 3,974 (25.34) | 1,066 (26.84) |

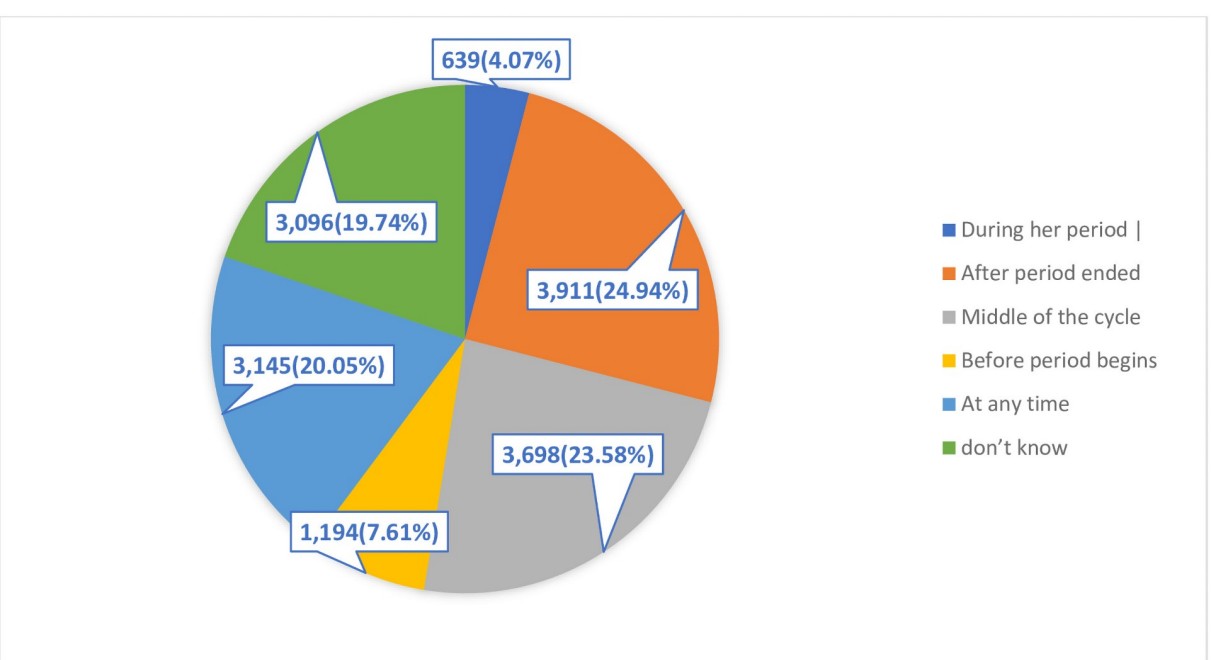

**Fig 1. A pie chart depicting the distribution of women with their knowledge of the ovulatory cycle.**

4.58) among those who had self-reported knowledge of contraceptive methods compared to those who did not know about contraceptives. The odds of KOC was 1.26 times (95% CI = 1.13–1.39) among women who currently used contraceptives relative to those who did not use. Women who attended higher education, secondary and primary education had 4.24 (95% CI = 3.54–5.07), 2.89 (95% CI = 2.48–3.36), and 1.57 (95% CI = 1.39–1.78) odds of KOC, respectively as compared to those who did not attend formal education (no education). Category of women from households with the richest, richer, middle, and poor wealth index level reported 1.71 (95% CI = 1.35–2.16), 1.42 (95% CI = 1.16–1.72), 1.29 (95% CI = 1.07–1.56), and 1.24 times (95% CI = 1.03–1.48) odds of KOC, respectively, in contrast to those who were from the poorest households. The likelihood of KOC was 1.13 (95% CI = 1.03–1.24) and 1.2 (95% CI = 1.07–1.35) times in women who menstruated in the last six weeks before the survey and those who had individual-level media exposure, respectively, compared to the references. Finally, from the community-level factors, only community-level media exposure was associated with KOC in that the odds of KOC was 1.53 times (95% CI = 1.24–1.88) among women from the area of high community-level media exposure compared to their counterparts.

We estimated the clustering effect because of the hierarchical nature of the DHS data. In the random-effects analysis, the ICC in the null model was high. This shows that about 25.15% of the variability of KOC was attributed to the difference between communities/clusters with the remaining 74.85% of the total variation was because of the individual variation. The higher MOR value in the null model revealed that there was a significant difference in KOC between communities/clusters. In the null model, the MOR value was 2.66 (95% CI: 2.514–2.951), depicting that if we witnessed two women from two different clusters, a woman in the cluster with high KOC was 2.66 times higher likelihood of having correct KOC as compared to a woman in the cluster with lower KOC. Also, model fitness was checked using deviance where the model with the lowest deviance was considered as the best fit model i.e. Model III in this case with a deviance of 14832.06 (Table 2).

**Table 2. Determinant factors of correct KOC among women of childbearing age in Ethiopia using multilevel logistic regression and the random-effects analysis (n = 16583).**

| Variables | | Null model | Model I AOR (95% CI) | Model II AOR (95% CI) | Model III AOR (95% CI) |
|---|---|---|---|---|---|
| Age in years | 15–19 | | 1 | - | 1 |
| | 20–24 | | 1.46 (1.28–1.68)*** | - | 1.46 (1.28–1.68)*** |
| | 25–29 | | 1.73 (1.50–1.99)*** | - | 1.72 (1.49–1.99)*** |
| | 30–34 | | 2.22 (1.90–2.59)*** | - | 2.21 (1.89–2.58)*** |
| | 35–39 | | 1.79 (1.52–2.11)*** | - | 1.78 (1.51–2.09)*** |
| | 40–44 | | 1.98 (1.66–2.38)*** | - | 1.97 (1.65–2.37)*** |
| | 45–49 | | 1.79 (1.46–2.21)*** | - | 1.78 (1.44–2.19)*** |
| Knowledge of contraceptives | No | | 1 | - | 1 |
| | Yes | | 3.16 (2.12–4.69)*** | - | 3.08 (2.07–4.58)*** |
| Educational status of women | No education | | 1 | - | 1 |
| | Primary | | 1.57 (1.39–1.77)*** | - | 1.57 (1.39–1.78) *** |
| | Secondary | | 2.88 (2.48–3.34)*** | - | 2.89 (2.48–3.36) *** |
| | Higher | | 4.26 (3.57–5.09)*** | - | 4.24 (3.54–5.07) *** |
| Wealth index | Poorest | | 1 | - | 1 |
| | Poorer | | 1.27 (1.07–1.52)** | - | 1.24 (1.03–1.48)* |
| | Middle | | 1.36 (1.13–1.63)*** | - | 1.29 (1.07–1.56)** |
| | Richer | | 1.53 (1.27–1.84)*** | - | 1.42 (1.16–1.72)*** |
| | Richest | | 2.17 (1.80–2.61)*** | - | 1.71 (1.35–2.16) *** |
| Current contraceptive use | No | | 1 | - | 1 |
| | Yes | | 1.26 (1.14–1.40)*** | - | 1.26 (1.13–1.39)*** |
| Menstruated in last six weeks | No | | 1 | - | 1 |
| | Yes | | 1.14 (1.04–1.24)** | - | 1.13 (1.03–1.24** |
| Women's media exposure | No | | 1 | - | 1 |
| | Yes | | 1.27 (1.13–1.42)*** | - | 1.20 (1.07–1.35)** |
| Residence | Urban | | - | 1.76 (1.41–2.20)*** | 1.17 (0.91–1.51) |
| | Rural | | - | 1 | 1 |
| Community education | Low | | - | 1 | 1 |
| | High | | - | 1.20 (0.99–1.44) | 0.83 (0.68–1.01) |
| Community poverty | Low | | - | 1 | 1 |
| | High | | - | 1.40 (1.11–1.71)*** | 1.04 (0.83–1.30) |
| Community media exposure | Low | | - | 1 | 1 |
| | High | | - | 1.87 (1.52–2.30)*** | 1.53 (1.24–1.88) *** |
| Parameter | | Null model (95% CI) | Model I (95% CI) | Model II (95% CI) | Model III (95% CI) |
| Community-level variance | | 1.106 (0.942–1.298) | 0.549 (0.453–0.666) | 0.550 (0.455–0.664) | 0.532 (0.438–0.646) |
| ICC (%) | | 25.15 (22.26–28.28) | 14.31 (12.10–16.84) | 14.32 (12.15–16.79) | 13.92 (11.75–16.42) |
| MOR | | 2.659 (2.514–2.951) | 2.022 (1.895–2.172) | 2.022 (1.898–2.169) | 1.999 (1.875–2.146) |
| PCV (%) | | 1 | 47.9655 | 50.3074 | 49.8207 |
| Log-likelihood | | -7878.95 | -7428.74 | -7729.60 | -7416.03 |
| LR test | | $X^2$ = 1376.13, p<0.001 | $X^2$ = 476.78, p< 0.001 | $X^2$ = 530.10, p< 0.001 | $X^2$ = 459.24, p< 0.001 |
| Deviance | | 15757.90 | 14857.48 | 15459.20 | 14832.06 |

*0.01 < p ≤ 0.05,

**0.001 < p ≤ 0.01,

***p ≤ 0.001.

## Discussion

Knowledge of the ovulatory cycle enables the women of childbearing age to plan for pregnancy or avoiding it. The current study aimed to identify the individual and community-level determinant factors of correct KOC among women of childbearing age in Ethiopia. In this study, correct KOC was reported by 23.58% of women in Ethiopia. It is similar to studies conducted in Kenya (23.4%), Gambia (23.1%), and Guinea (23.3%) [11]. The result is higher than other studies in India (15%) [10], and African countries (like Sao Tome and Principe (10.4%), Namibia (13.9%), Nigeria (20.3%), Zambia (21.5%), and Rwanda (21%)) [11]. However, the prevalence found in this study is lower than other studies in African countries (Comoros (49%), Togo (42.8%), Ghana (34%), and Sierra Leone (30.3%)) [11], and the USA (32.8%) [12]. These discrepancies might be because of differences in socioeconomic status, and sociocultural variations between women of childbearing age of these countries.

In the multilevel multivariable analysis, after considering the community level variability, age of women, knowledge of contraceptives, educational status of women, wealth index of the household, current use of contraceptives, menstruating in the last six weeks, women's media exposure, and community level media exposure were determinants of correct KOC.

After adjusting for other factors, women's age had a statistically significant association with correct KOC in which women in the advanced age category were more likely to have a correct KOC compared to teenagers (15–19 years). This is supported by a study in Ghana [16] and the USA [15]. A study conducted in Ethiopia using EDHS also supported our finding even though that study used simple logistic regression analysis without adjusting for community-level factors [20]. The reason for this association could be as age is increased, exposure to different reproductive related issues is increased that lets women gain more knowledge. Increased educational status of women was associated with KOC, which is supported by other studies [14,17–19,26]. This explicates that education has a positive impact on health knowledge and behaviour [27]. Women from the richest, richer, middle, and poor households had better correct KOC than those who were from the poorest households. This is supported by another study in Africa [14]. This might be because women with a higher wealth index are more likely to be knowledgeable [28] and have the interest to learn [29].

Self-reported knowledge of contraceptive and the current use of contraceptive methods were also determinants for correct KOC, which is in line with other studies [16,30]. This might be because counselling on contraceptive use increases fertility knowledge [31] and knowledge of ovulation is included in family planning counselling guidelines [32]. Women who menstruated in the last six weeks had 1.13 times KOC compared to their counterparts. There is no literature supporting or against this association, but it might be because women who observed menstruation in the recent weeks are more likely to better understand the time of ovulation than those who did not. Women who did not report menstruation, being a sign of pregnancy, may be an unintended pregnancy, partly because of their low KOC.

Individual-level and community-level media exposure were determinants of KOC. Mass media exposure has a positive impact on comprehensive knowledge [33]. Besides, community-level media exposure increases maternal health service utilization, which is the major way of transmitting health information including family planning such as knowledge of reproductive cycle (ovulation) [34]. The findings of this study suggests the Ethiopian government, particularly the Ministry of health, for designing strategies to increase KOC and other reproductive aspects of women's health to achieve the good health goal which is one of the 2030 agenda of sustainable development goals [35].

## Strength and limitations of the study

This study was based on nationally representative data with appropriate statistical analysis (weighting and multilevel analysis). However, the interpretations of this study could be made by considering the following limitations; we used secondary data, which missed a few important variables, such as pre-survey awareness of the ovulatory cycle among women of childbearing age, for the analysis. Because of the cross-sectional nature, it cannot infer the cause-effect relationship between factors and outcome variables. Social desirability and recall biases were also the expected limitations of this study.

## Conclusions

Correct KOC was low, which demands the need to design and implement reproductive health services via community media campaign and health promotion. The investigators of this study would like to recommend the Ethiopian Ministry of Health to focus on the reproductive health of women by giving privilege to the teenagers, those with lower educational level, and lower socioeconomic status. Also, the strengthening of media campaigns could increase women's correct KOC, which is crucial for precluding unintended pregnancy. Besides, we urge the Ministry of Health, Ministry of Science and Higher education, and the Universities to institute reproductive health promotion projects, including the above factors. We also advise the forthcoming researchers to conduct the level of awareness of women of childbearing age about the ovulatory cycle so that awareness creation can be tailored based on the findings.

## Acknowledgments

The authors acknowledged the measures DHS program for providing the dataset and for the permission to do this research.

## Author Contributions

**Conceptualization:** Baye Dagnew, Achamyeleh Birhanu Teshale, Henok Dagne, Mengistie Diress, Getayeneh Antehunegn Tesema, Reta Dewau, Meseret Derbew Molla, Yigizie Yeshaw.

**Data curation:** Baye Dagnew, Achamyeleh Birhanu Teshale, Henok Dagne, Mengistie Diress, Getayeneh Antehunegn Tesema, Reta Dewau, Meseret Derbew Molla, Yigizie Yeshaw.

**Formal analysis:** Baye Dagnew.

**Methodology:** Baye Dagnew, Achamyeleh Birhanu Teshale, Henok Dagne, Mengistie Diress, Getayeneh Antehunegn Tesema, Reta Dewau, Meseret Derbew Molla, Yigizie Yeshaw.

**Resources:** Baye Dagnew.

**Software:** Baye Dagnew.

**Visualization:** Baye Dagnew, Achamyeleh Birhanu Teshale, Henok Dagne, Mengistie Diress, Getayeneh Antehunegn Tesema, Reta Dewau, Meseret Derbew Molla, Yigizie Yeshaw.

**Writing – original draft:** Baye Dagnew.

**Writing – review & editing:** Baye Dagnew, Achamyeleh Birhanu Teshale, Henok Dagne, Mengistie Diress, Getayeneh Antehunegn Tesema, Reta Dewau, Meseret Derbew Molla, Yigizie Yeshaw.

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
