## [Decision Letter · Decision Letter 0]

24 Mar 2021

PONE-D-20-30888

Individual and community-level determinants of knowledge of ovulatory cycle among women of childbearing age in Ethiopia: A Multilevel analysis based on the 2016 Ethiopian Demographic and Health Survey

PLOS ONE

Dear Dr. Dagnew,

Thank you for submitting your manuscript to PLOS ONE. After careful consideration, we feel that it has merit but does not fully meet PLOS ONE’s publication criteria as it currently stands. Therefore, we invite you to submit a revised version of the manuscript that addresses the points raised during the review process.

Please pay particular attention to providing the methodological clarifications requested by the reviewers, and addressing their recommendations regarding the presentation and interpretation of the data.

We look forward to receiving your revised manuscript.

Kind regards,

Jamie Males

Senior Editor

PLOS ONE

Journal Requirements:

2. In the methods section, please provide a statement as to whether permission was granted from the DHS prior to obtaining the data.

Reviewers' comments:

Reviewer's Responses to Questions

**Comments to the Author**

1. Is the manuscript technically sound, and do the data support the conclusions?

Reviewer #1: Yes

Reviewer #2: Partly

2. Has the statistical analysis been performed appropriately and rigorously? 

Reviewer #1: Yes

Reviewer #2: Yes

3. Have the authors made all data underlying the findings in their manuscript fully available?

Reviewer #1: Yes

Reviewer #2: No

4. Is the manuscript presented in an intelligible fashion and written in standard English?

Reviewer #1: No

Reviewer #2: No

5. Review Comments to the Author

Reviewer #1: Introduction:

*Please elaborate a bit further on the previous studies conducted on KOC in Africa. What type of studies have been conducted and what were their results?

* Please spell out EDHS at first use.

* Please motivate why determining factors associated with KOC would be important, what implications such knowledge might have to improve womens' reproductive health.

Method:

* Please define the range of what was determined to be the "childbearing age", i.e. the study population.

* Please motivate why p < 0.2 was used to select variables for inclusion in one of part the analyses and p < 0.05 was used to select variables for inclusion in another part of the analysis.

* Please motivate why menstruation within the past 6 weeks was included as one of the variables under investigation, i.e. how this variable might have an association with KOC.

Results:

* Please state the number and % of women with missing values on the outcome variable (KOC).

* Please include CI for the ORs mentioned in the text.

* Figure 1: Please consider if a pie chart might be more appropriate in order to present distribution of responses.

Discussion:

* Limitations: Please specify what important variables you consider were missing for the analysis.

Conclusion:

* Please specify how you consider that the health section should promote reproductive health among the subgroups of women identified in the study.

Pleas note that the manuscript needs considerable language revision before publication.

Reviewer #2: This paper addresses a very important reproductive health subject and will potentially contribute to the literature in this area. However, I have some comments that will help to significantly improve the paper.

Title:

Okay

Abstract

This section is quite okay.

Introduction

While this section is quite okay, authors should also provide evidence of KOC in Europe as provided in America and Asia (India).

Authors should also provide a brief explanation of the significance of the study after the objective statement.

Materials and Methods

Under the data source, authors should put the link of the data source in a bracket. Also, provide some more information on the data including the demographic profile (sex and age) of the participants of the survey. Who conducted the survey and who are the collaborators?

Under population, even though this may be known, provide the age range instead of saying childbearing age.

Under the sampling procedure, the authors should provide a few more details. For instance, what is the total number of women that took part in the survey, and what is the total number of listed households from which the fixed 28 were selected.

Under study variables, authors should provide a better description of the outcome variable. For instance, after describing the nature of the original outcome variable, what is the nature of the final outcome variable and its measurement?

Authors should explain whether there are re-codings or use of original variables and how they were measured.

The authors did not state the level of the community factors. Is it EAs or clusters or others?

Were the independent variables selected arbitrarily or informed by existing literature? If former, explain, and if the latter, provide citations to support the selection of variables.

It is not clear how the authors derived the community-level variables from the individual-level factors. More details should be provided.

Under the statistical analysis, the authors stated that they executed women's sampling weight, however, this is not enough as authors should apply survey design due to the hierarchical nature of the data.

Results

It is better for authors to delete “contraceptive profiles” from the title of the descriptive results.

Authors should connect the descriptive title with the Table 1 title.

Under Knowledge of the ovulatory cycle, 24.94% cannot be said as the majority, so the statement should be revised.

It is more understandable and convenient to have both the fixed and random effects in the same table. Therefore, authors should merge Tables 2 and 3 and synchronized them.

The correct KOC results in Table 3 are descriptive and cannot be presented with the model results in the same table. Authors should consider merging them with Table 1 or creating a new table for them.

Authors should revise the “due to attributed to” in the second sentence under random effects.

Authors should avoid the use of for instance, 3 times better than, in reporting the odds ratios. It is appropriate to say the odds of KOC were 3 times compared to the reference group. This cuts across the fixed effects results and should be revised as such.

Discussion

The authors should explain how sample size could determine differences in KOC prevalence among countries or remove it. I do not find it convincing.

Age in “Age of women” should be all small letters.

Women ages 15-19 years are not young adults as stated under the multivariate analysis discussion but teenagers and should be revised. Also, authors should revise the “comfortable wealth index” to a higher wealth index.

The explanation for menstruation looks unconvincing. The variable is about having or not having menstruation in the past 6 weeks but not about remembering their menstruation. What about those not having menstruation being a sign of pregnancy which may be an unintended pregnancy partly due to their low KOC as shown by the odds ratio.

Conclusion

The first sentence does not read well, revise. It is better to say KOC is low rather than very low.

The “health sector” is not a decision and policymaker and therefore, authors should state the agencies directly involved in this.

General comments

It appears the authors did not follow the journal formats by not providing line numbers and making referring difficult. The required square bracket for in-text citation was replaced with a round bracket among several others including font size of titles. All the sections after the conclusion to the author’s contribution do not fit the journal format. The authors should visit the journal format documents and use them to format the manuscript appropriately. Also, the references have different font and font size from the main manuscript. Finally, the manuscript should be proofread and revised as some of the sentences are not clear enough or include typos and are found under the “Knowledge of the ovulatory cycle” section.

6. PLOS authors have the option to publish the peer review history of their article (what does this mean?). If published, this will include your full peer review and any attached files.

Reviewer #1: No

Reviewer #2: No

---

## [Author Response · Author response to Decision Letter 0]

2 Apr 2021

Point-by-point response to reviewers

We, the authors, would like to acknowledge the editorial team and the reviewers for giving us the opportunity to revise and improve our manuscript titled “Individual and community-level determinants of knowledge of ovulatory cycle among women of childbearing age in Ethiopia: A Multilevel analysis based on the 2016 Ethiopian Demographic and Health Survey” and ID “PONE-D-20-30888”. We addressed all the concerns of the reviewers and the responses are stated by the point-by-point response letter (thee authors’ replies are italic-bold) as well as corrections are incorporated in the revised manuscript. 

Authors’ reply to the comments of the Reviewer #1 

Reviewer’s comment: Please elaborate a bit further on the previous studies conducted on KOC in Africa. What type of studies has been conducted and what were their results?

Authors’ reply: Thank you. We included few studies conducted in Africa as in Sub-Saharan Africa and also a study in Ghana and found in the revised manuscript. Besides, most studies in Africa are described in the discussion section to compare our findings (Line#68-70). 

Reviewer’s comment: Please spell out EDHS at first use. 

Authors’ reply: Thank you. We spelled out as suggested (Line#77).

Reviewer’s comment: Please motivate why determining factors associated with KOC would be important, what implications such knowledge might have to improve women’s' reproductive health.

Authors’ reply: Thank you. The results of this study will be helpful to institute reproductive health specific interventional strategies to equip childbearing age women with adequate and appropriate knowledge of the ovulatory cycle. Correct KOC is important to improve women’s health by preventing unintended pregnancy especially for those with poor access and awareness of contraceptive methods particularly in Sub-Saharan Africa. Identifying the determinant factors is used to target the associated factors that lead women for correct KOC so that strategies will be designed to improve positively associated factors and to prevent the negatively influencing factors. All these are included in the revised manuscript(Line#78-85).

Reviewer’s comment: Please define the range of what was determined to be the "childbearing age", i.e. the study population.

Authors’ reply: Thank you very much for the compliment. We added the range to illustrate the childbearing age as “15-49 years old” (Line#115).

Reviewer’s comment: Please motivate why p < 0.2 was used to select variables for inclusion in one of part the analyses and p < 0.05 was used to select variables for inclusion in another part of the analysis.

Authors’ reply: Thank you. As it is known, in the scientific community variables for the selection of eligible variables for the multivariable analysis we used p-value <0.20 and this is the usual procedure to drop variables that are distal or unnecessary for the data at hand. After we have selected variables using p <0.20 we conducted the multivariable analysis. In the multivariable analysis, we used p value<0.05 for declaring variables as predictors of KOC. The reason why we used p<0.05 is just to minimize the association due to chance. 

Reviewer’s comment: Please motivate why menstruation within the past 6 weeks was included as one of the variables under investigation, i.e. how this variable might have an association with KOC.

Authors’ reply: Thank you. We included the variable “menstruation within the past two weeks” in a sense that women who recently practiced menstruation (last menstruation prior to the survey) may have a relatively better knowledge of ovulatory cycle; they are more likely to search out why menstruation occurred (as they observed it), when to occur, and the causes of ovulation. As it is known, the female reproductive cycle comprises two cycle; the ovarian cycle (including the ovulation phase) and uterine cycle (including menstrual phase). After ovulation, if there is no fertilization, there will be withdrawal of female gonadal hormones that results in menstruation after successive event. Therefore, females who observed menstruation are more likely to understand the time of ovulation than those who did not (may be forgotten as it becomes longer). That is why we included this variable as potential exposure variable for the outcome variable (Line#288-90). 

Reviewer’s comment: Please state the number and % of women with missing values on the outcome variable (KOC).

Authors’ reply: Thank you. All the women who took part in the survey (15,683) responded to the outcome variable (KOC). Therefore, there is no missing case for the KOC (Line#127-8).

Reviewer’s comment: Please include CI for the ORs mentioned in the text.

Authors’ reply: Thank you for the compliment. We included the 95% CI for each OR mentioned in the text (Line#222-38).

Reviewer’s comment: Figure 1: Please consider if a pie chart might be more appropriate in order to present distribution of responses

Authors’ reply: Thank you. We used pie chart instead of bar chart as suggested in the revised manuscript (separate figure uploaded). 

Reviewer’s comment: Please specify what important variables you consider were missing for the analysis.

Authors’ reply: Thank you. We think all the important variables are included in the survey. However, if pre-survey “awareness” about ovulatory cycle is included, it will be fine. Awareness about something is the main way of knowledge gain (Line#304).

Reviewer’s comment: Please specify how you consider that the health section should promote reproductive health among the subgroups of women identified in the study.

Authors’ reply: Thank you. As per our understanding, we urge the Ministry of health, Ministry of science higher education, and the Universities to institute reproductive health promotion projects including the above-mentioned factors. We also advice the forthcoming researchers to conduct the level of awareness of childbearing aged women about ovulatory cycle so that awareness creation can be tailored based on the finding (Line#311-8). 

Reviewer’s comment: Please note that the manuscript needs considerable language revision before publication.

Authors’ reply: Thank you very much. We revised the whole manuscript for language revision and we believe we improved the language problems. 

Reviewer#2

Reviewer’s comment: While this section is quite okay, authors should also provide evidence of KOC in Europe as provided in America and Asia (India). 

Author’s reply: Thank you. As far as our searching strategy, there is scarce data in Europe regarding the proportion of childbearing age women on knowledge of ovulation cycle. But, we found one study in Spain and we included it in the revised manuscript.

Reviewer’s comment: Authors should also provide a brief explanation of the significance of the study after the objective statement.

Author’s reply: Thank you. We included further explanation of the significance of the study after the objective as suggested (Line#77-85).

Reviewer’s comment: Under the data source, authors should put the link of the data source in a bracket. Also, provide some more information on the data including the demographic profile (sex and age) of the participants of the survey. Who conducted the survey and who are the collaborators?

Authors’ reply: Thank you. We put the link in bracket as suggested (Line#88). WE used the Ethiopia Population and Housing Census (2007) as a sampling frame for the EDHS 2016. The census frame is a complete list of 84,915 enumeration areas (EAs) created for the 2007 PHC. An EA is a geographic area covering on average 181 households (~15,369,615 household on average). Each large EA was segmented to minimize the task of household listing from which only one segment was selected for the survey with probability proportional to segment size. Household listing was conducted only in the selected segment; that is, a 2016 EDHS cluster is either an EA or a segment of an EA. Then, a total of 18,008 households were selected for the sample, of which 17,067 were occupied. Of the occupied households, 16,650 were successfully interviewed, yielding a response rate of ~98%. Regarding the participants, 15,683 (15-49 years of age) were females and 12,688 (15-59 years of age) were males. Regarding the organizations involved in the survey, the Central Statistical Agency (CSA) conducted the survey in collaboration with the Ethiopian Ministry of Health and Ethiopian Public health institute and also the international classification of functioning, disability and health (ICF) involved in technical assistance. The financial support was obtained from the development partners i.e. the United States Agency for International Development (USAID), the government of the Netherlands, the Global Fund, the Irish Aid, the World Bank, the United Nations Population Fund (UNFPA), the United Nations Children’s Fund (UNICEF), and UN Women. The dataset, we used for this analysis, comprises all females aged 15-49 years. This is included in the revised manuscript (Line#87-106).

Reviewer’s comment: Under population, even though this may be known, provide the age range instead of saying childbearing age.

Authors’ reply: Thank you. We included the age range (15-49 years old) as suggested (Line#115).

Reviewer’s comment: Under the sampling procedure, the authors should provide a few more details. For instance, what is the total number of women that took part in the survey, and what is the total number of listed households from which the fixed 28 were selected.

Authors’ reply: Thank you. ~15,369,615 household on average. A total of 15,683 women took part in the survey (all of which responded to the outcome variable KOC) (Line#126-8). 

Reviewer’s comment: Under study variables, authors should provide a better description of the outcome variable. For instance, after describing the nature of the original outcome variable, what is the nature of the final outcome variable and its measurement? Authors should explain whether there are re-codings or use of original variables and how they were measured.

Authors’ reply: Thank you. We added the suggested comments in the revised manuscript… We performed recoding of the variables so that the outcome variable become binary (dichotomized) which is useful to execute regression. For this, we created a new variable called correct knowledge of the ovulatory cycle. The alternative “middle of the cycle was coded as 1” and others were coded as 0” (Line#137-44). 

Reviewer’s comment: The authors did not state the level of the community factors. Is it EAs or clusters or others?

Authors’ reply: Thank you. When we say community level factors, we mean factors created at cluster level using their respective individual level factors to assess the effect of these variables at community or cluster level and these are stated in the revised manuscript (Line#144-6). 

Reviewer’s comment: Were the independent variables selected arbitrarily or informed by existing literature? If former, explain, and if the latter, provide citations to support the selection of variables.

Authors’ reply: We used three approaches to select variables. These were based on the clinical plausibility of variables, based on the existing literature (cited in the background section; Line#70-2)), and based on the variables found in the EDHS dataset. We performed univariate analysis one by one with the outcome variable (KOC) based on the fact that each variable has biological or other association with the outcome variable. In the univariate analysis, those variables with a p<0.2, were selected and entered into multivariable multilevel logistic regression. In the final model, we used p<0.05 to declare statistical significance. 

Reviewer’s comment: It is not clear how the authors derived the community-level variables from the individual-level factors. More details should be provided.

Authors’ reply: Thank you very much. We described more detail how the community-level factors are create and amended in the revised manuscript (Line#147-59). 

Reviewer’s comment: Under the statistical analysis, the authors stated that they executed women's sampling weight, however, this is not enough as authors should apply survey design due to the hierarchical nature of the data.

Authors’ reply: We have weighted the data using women sample weighting, primary sampling unit, and stratification variable (to account for survey design) and weighting to account for representativeness, non-response rate, and to get appropriate statistical estimate (Line#162-6).

Reviewer’s comment: It is better for authors to delete “contraceptive profiles” from the title of the descriptive results.

Authors reply: Thank you. We deleted contraceptive profiles as suggested (Line#)

Reviewer’s comment: Authors should connect the descriptive title with the Table 1 title

Authors’ reply: We made correction as suggested (Line#).

Reviewer’s comment: Under Knowledge of the ovulatory cycle, 24.94% cannot be said as the majority, so the statement should be revised.

Authors’ reply: Thank you for the compliment. We did as suggested (Line#208). 

Reviewer’s comment: It is more understandable and convenient to have both the fixed and random effects in the same table. Therefore, authors should merge Tables 2 and 3 and synchronized them.

Authors’ reply: Thank you we merged table 2 and 3 as suggested. 

Reviewer’s comment: The correct KOC results in Table 3 are descriptive and cannot be presented with the model results in the same table. Authors should consider merging them with Table 1 or creating a new table for them.

Authors’ reply: Thank you very much. We merged the results of KOC with table 1 as suggested. 

Reviewer’s comment: Authors should revise the “due to attributed to” in the second sentence under random effects. Authors should avoid the use of for instance, 3 times better than, in reporting the odds ratios. It is appropriate to say the odds of KOC were 3 times compared to the reference group. This cuts across the fixed effects results and should be revised as such.

Authors’ reply: Thank you. Thank you. We removed “due to” which is grammatical error (Line#242, and Line#222-239)

Reviewer’s comment: The authors should explain how sample size could determine differences in KOC prevalence among countries or remove it. I do not find it convincing.

Authors’ reply: Thank you. We removed sample size as suggested (Line#266).

Reviewer’s comment: Age in “Age of women” should be all small letters.

Authors’ reply: Thank you. We make it small letter as suggested (Line#268). 

Reviewer’s comment: Women ages 15-19 years are not young adults as stated under the multivariate analysis discussion but teenagers and should be revised. Also, authors should revise the “comfortable wealth index” to a higher wealth index.

Authors’ reply: Thank you. We revised as suggested (Line#274=teenagers, and Line#281 higher). 

Reviewer’s comment: The explanation for menstruation looks unconvincing. The variable is about having or not having menstruation in the past 6 weeks but not about remembering their menstruation. What about those not having menstruation being a sign of pregnancy which may be an unintended pregnancy partly due to their low KOC as shown by the odds ratio.

Authors’ reply: The female reproductive cycle comprises two cycles; the ovarian cycle (the ovulation is one of the three phases) and uterine cycle (menstrual phase is one of the three phases). After ovulation, if there is no fertilization, there will be withdrawal of female gonadal hormones (estrogen and progesterone) that results in menstruation after successive event. Therefore, females who observed menstruation are more likely to understand the time of ovulation than those who did not (may be forgotten as it becomes longer). We also included the reason suggested for unintended pregnancy (if we capture the comment in the right way (Line#287-91). 

Reviewer’s comment: The first sentence does not read well, revise. It is better to say KOC is low rather than very low. The “health sector” is not a decision and policymaker and therefore, authors should state the agencies directly involved in this.

Authors’ reply: Thank you. We corrected as suggested in the revised manuscript (Line# 352, we say it low as suggested). Line#311, 314, and 315 we included Ethiopian ministry of health, Ministry of Science and Higher education, and the Universities). 

Reviewer’s comment: It appears the authors did not follow the journal formats by not providing line numbers and making referring difficult. The required square bracket for in-text citation was replaced with a round bracket among several others including font size of titles. All the sections after the conclusion to the author’s contribution do not fit the journal format. The authors should visit the journal format documents and use them to format the manuscript appropriately. Also, the references have different font and font size from the main manuscript. Finally, the manuscript should be proofread and revised as some of the sentences are not clear enough or include typos and are found under the “Knowledge of the ovulatory cycle” section.

Authors’ reply: We made square bracket for in-text citation, we included line number, we corrected the font sizes as 18 for title pages (first heading), 16 for second heading, 14 for 3rd heading and 12 for the body of the text. We corrected the format using the journals guideline, corrected the references font size and style as suggested, and finally we made revisions on typos and grammatical errors throughout the manuscript particularly the suggested section.

---

## [Decision Letter · Decision Letter 1]

28 Apr 2021

PONE-D-20-30888R1

Individual and community-level determinants of knowledge of ovulatory cycle among women of childbearing age in Ethiopia: A Multilevel analysis based on 2016 Ethiopian Demographic and Health Survey

PLOS ONE

Dear Dr. Dagnew,

Thank you for submitting your manuscript to PLOS ONE. After careful consideration, we feel that it has merit but does not fully meet PLOS ONE’s publication criteria as it currently stands. Therefore, we invite you to submit a revised version of the manuscript that addresses the points raised during the review process.

There has been a change of editor since the last revision, and the two previous reviewers where reassigned for review before the new editor was assigned. Except for minor reservations, they feel that suggested changes were incorporated.

However, upon initial assesment by the new editor, I have to note with much regret that this submission carries out almost identical analysis with the same data as another paper that had been published before the initial submission. Note that PLOS ONE does not accept for publication studies that have already been published elsewhere in the peer-reviewed literature and if a submitted study replicates or is very similar to previous work, authors must provide a sound scientific rationale for the submitted work and clearly reference and discuss the existing literature. Submissions that replicate or are derivative of existing work will likely be rejected if authors do not provide adequate justification.

I know this should have been identified in the previous round and I apologize for the late feedback in that respect. 

Reference:

Getahun MB, Nigatu AG. Knowledge of the Ovulatory Period and Associated Factors Among Reproductive Women in Ethiopia: A Population-Based Study Using the 2016 Ethiopian Demographic Health Survey. *Int J Womens Health*. 2020;12:701-707

https://doi.org/10.2147/IJWH.S267675

You must discuss this article among the literature review and incorporate its findings. If you can make a convincing case of why the article should be considered despite this, the article would be evaluated further.

We look forward to receiving your revised manuscript.

Kind regards,

José Antonio Ortega, Ph.D.

Academic Editor

PLOS ONE

Reviewers' comments:

Reviewer's Responses to Questions

**Comments to the Author**

1. If the authors have adequately addressed your comments raised in a previous round of review and you feel that this manuscript is now acceptable for publication, you may indicate that here to bypass the “Comments to the Author” section, enter your conflict of interest statement in the “Confidential to Editor” section, and submit your "Accept" recommendation.

Reviewer #1: All comments have been addressed

Reviewer #2: (No Response)

2. Is the manuscript technically sound, and do the data support the conclusions?

Reviewer #1: Yes

Reviewer #2: Yes

3. Has the statistical analysis been performed appropriately and rigorously? 

Reviewer #1: Yes

Reviewer #2: Yes

4. Have the authors made all data underlying the findings in their manuscript fully available?

Reviewer #1: Yes

Reviewer #2: No

5. Is the manuscript presented in an intelligible fashion and written in standard English?

Reviewer #1: Yes

Reviewer #2: No

6. Review Comments to the Author

Reviewer #1: (No Response)

Reviewer #2: The authors have effectively addressed the majority of the comments provided by reviewers. However, a few comments were only responded to by the authors without effecting the needed revisions in the manuscript.

1. Authors stated that data used for the study were included in the manuscript but were not actually included. Authors should provided the link to the dhsprogram data for EDHS 2016 in the editorial manager system in place of the "All relevant data are included in the manuscript" statement.

2. The existing literature supporting the variables was only explained but not provided as recommended. Some of these citations should be provided in the Study Variables section to support the variable selection.

3. The first paragraph of the "Study variables" has a different spacing from the rest of the manuscript.

4. The "BMC Series" format of manuscript presented after the conclusion section should be removed (List of abbreviations to author contributions, EXCEPT Acknowledgements) as all these are pre-filled in and generated from the editorial manager system onto the final paper.

5. Some clear grammatical problems remain in the manuscript and authors have to submit the manuscript for expert copy-editing before re-submission.

7. PLOS authors have the option to publish the peer review history of their article (what does this mean?). If published, this will include your full peer review and any attached files.

Reviewer #1: **Yes: **Helena Litorp

Reviewer #2: No

---

## [Author Response · Author response to Decision Letter 1]

22 May 2021

Point-by-point response to the editor and reviewers

Academic editor’s comment: I have to note with much regret that this submission carries out almost identical analysis with the same data as another paper that had been published before the initial submission. Note that PLOS ONE does not accept for publication studies that have already been published elsewhere in the peer-reviewed literature and if a submitted study replicates or is very similar to previous work, authors must provide a sound scientific rationale for the submitted work and clearly reference and discuss the existing literature. Submissions that replicate or are derivative of existing work will likely be rejected if authors do not provide adequate justification. I know this should have been identified in the previous round and I apologize for the late feedback in that respect. Ref: https://doi.org/10.2147/IJWH.S267675 . You must discuss this article among the literature review and incorporate its findings. If you can make a convincing case of why the article should be considered despite this, the article would be evaluated further.

Authors’ reply: The previous study used the Ethiopian Demographic and Health Survey (EDHS) 2016 [20]. The previous study did not consider the community level factors (they consider factors only at the individual-level; even they did not use multilevel analysis which is appropriate for hierarchical data), there was no weighting to account for appropriate estimation, and there were missing of essential variables such as media exposure which plays a crucial role for the provision of information including reproductive health. It is useful to consider the community-level factors for designing interventional strategies not only based from the individual-level factors. In a summary, our study differs from the previous one by considering the following;

1. Our study used sampling weight (using women's sampling weight) to account for the non-proportional allocation of the sample in each region and the potential variations in the response rates, as well as to get an appropriate statistical estimate. Besides, stratification was used to account for the survey design 

2. We included community-level factors (community-education, community poverty and community media exposure). As explained above, strategies should based from community level factors besides individual level factors. 

3. We used multilevel multivariable analysis; clustering effect was performed due to the hierarchical nature of the DHS data. Intra-class correlation coefficient (ICC) and median odds ratio (MOR) were computed to choose appropriate model. We fitted four models; null model/ (containing only the outcome variable), the model I (with individual-level factors only), model II (with community-level factors only), and model III (with the individual- and community-level factors simultaneously). Finally, we used the third model by looking the lowest deviance.

4. In addition to the individual factors included by the previous study, we included media exposure which is crucial factor for the provision of information including reproductive health knowledge. 

For more information, we included it in the revised manuscript Line#70-6 and in the discussion section 

Reviewers’ comments to the Authors

Authors’ reply to the reviewer#1: We acknowledge the reviewer for accepting our previous replies to the comments. In this phase, there is no comment from the reviewer#1 to be addressed.

Authors’ reply to the reviewer#2

Reviewer #2 (comments): The authors have effectively addressed the majority of the comments provided by reviewers. However, a few comments were only responded to by the authors without effecting the needed revisions in the manuscript.

1. Authors stated that data used for the study were included in the manuscript but were not actually included. Authors should provide the link to the dhsprogram data for EDHS 2016 in the editorial manager system in place of the "All relevant data are included in the manuscript" statement.

Authors’ reply: We provided the link to DHS program in the editorial system instead of “"All relevant data are included in the manuscript" statement. All the data were included in the manuscript. There is no restriction. The data can be accessed from the DHS website after reasonable request (https://dhsprogram.com/Data/terms-of-use.cfm)

2. The existing literature supporting the variables was only explained but not provided as recommended. Some of these citations should be provided in the Study Variables section to support the variable selection.

Authors’ reply: Thank you very much. We included few citations to the variables section to indicate where we get the variables to consider them as independent (Line#139).

3. The first paragraph of the "Study variables" has a different spacing from the rest of the manuscript.

Authors’ reply: Thank you. We corrected as suggested (formerly it was 1.5 and now we make it double spacing). 

4. The "BMC Series" format of manuscript presented after the conclusion section should be removed (List of abbreviations to author contributions, EXCEPT Acknowledgements) as all these are pre-filled in and generated from the editorial manager system onto the final paper.

Authors’ reply: Thank you. We removed as advised. 

5. Some clear grammatical problems remain in the manuscript and authors have to submit the manuscript for expert copy-editing before re-submission.

Authors’ reply: We invited an expert person from the department of literature at the University of Gondar for the revision of the grammatical and standard use of English language. Then, we incorporated the points from the invited person and all the authors revised the manuscript for the correction of grammatical flaws. We hope, we have made improvements in the revised manuscript.

---

## [Decision Letter · Decision Letter 2]

21 Jun 2021

Individual and community-level determinants of knowledge of ovulatory cycle among women of childbearing age in Ethiopia: A Multilevel analysis based on 2016 Ethiopian Demographic and Health Survey

PONE-D-20-30888R2

Dear Dr. Dagnew,

We’re pleased to inform you that your manuscript has been judged scientifically suitable for publication and will be formally accepted for publication once it meets all outstanding technical requirements.

Both referees feel that their comments have been properly addressed, and the main concern of the editor, providing a rationale for a study with the same topic and data than previous studies, has also been addressed.

Kind regards,

José Antonio Ortega, Ph.D.

Academic Editor

PLOS ONE

Additional Editor Comments (optional):

Reviewers' comments:

Reviewer's Responses to Questions

**Comments to the Author**

1. If the authors have adequately addressed your comments raised in a previous round of review and you feel that this manuscript is now acceptable for publication, you may indicate that here to bypass the “Comments to the Author” section, enter your conflict of interest statement in the “Confidential to Editor” section, and submit your "Accept" recommendation.

Reviewer #2: All comments have been addressed

2. Is the manuscript technically sound, and do the data support the conclusions?

Reviewer #2: Yes

3. Has the statistical analysis been performed appropriately and rigorously? 

Reviewer #2: Yes

4. Have the authors made all data underlying the findings in their manuscript fully available?

Reviewer #2: Yes

5. Is the manuscript presented in an intelligible fashion and written in standard English?

Reviewer #2: Yes

6. Review Comments to the Author

Reviewer #2: (No Response)

7. PLOS authors have the option to publish the peer review history of their article (what does this mean?). If published, this will include your full peer review and any attached files.

Reviewer #2: No

---

## [Editor Report · Acceptance letter]

24 Aug 2021

PONE-D-20-30888R2 

Individual and community-level determinants of knowledge of ovulatory cycle among women of childbearing age in Ethiopia: A Multilevel analysis based on 2016 Ethiopian Demographic and Health Survey 

Dear Dr. Dagnew:

I'm pleased to inform you that your manuscript has been deemed suitable for publication in PLOS ONE. Congratulations! Your manuscript is now with our production department. 

Kind regards, 

on behalf of

Dr. José Antonio Ortega 

Academic Editor

PLOS ONE